# Characterization of a Programmable Argonaute Nuclease from the Mesophilic Bacterium *Rummeliibacillus suwonensis*

**DOI:** 10.3390/biom12030355

**Published:** 2022-02-23

**Authors:** Xiaoman Jiang, Yang Liu, Qi Liu, Lixin Ma

**Affiliations:** State Key Laboratory of Biocatalysis and Enzyme Engineering, Hubei Collaborative Innovation Center for Green Transformation of Bio-Resources, Hubei Key Laboratory of Industrial Biotechnology, School of Life Sciences, Hubei University, Wuhan 430062, China; xm_jiang520@163.com (X.J.); lyang@stu.hubu.edu.cn (Y.L.); liuqiicloud@163.com (Q.L.)

**Keywords:** argonaute, nuclease, mesophilic, DNA cleavage, RNA cleavage

## Abstract

Prokaryotic Argonautes (pAgos) from mesophilic bacteria are attracting increasing attention for their genome editing potential. So far, it has been reported that KmAgo from *Kurthia massiliensis* can utilize DNA and RNA guide of any sequence to effectively cleave DNA and RNA targets. Here we find that three active pAgos, which have about 50% sequence identity with KmAgo, possess typical DNA-guided DNA target cleavage ability. Among them, RsuAgo from *Rummeliibacillus suwonensis* is mainly explored for which can cleave both DNA and RNA targets. Interestingly, RsuAgo-mediated RNA target cleavage occurs only with short guide DNAs in a narrow length range (16–20 nt), and mismatches between the guide and target sequence greatly affect the efficiency of RNA target cleavage. RsuAgo-mediated target cleavage shows a preference for a guide strand with a 5′-terminal A residue. Furthermore, we have found that RsuAgo can cleave double-stranded DNA in a low-salt buffer at 37 °C. These properties of RsuAgo provide a new tool for DNA and RNA manipulation at moderate temperatures.

## 1. Introduction

Eukaryotic Argonautes (eAgos) are key players in all small-RNA-guided gene-silencing processes, which not only have the function of loading small RNA, but also play an important role in transcription, alternative splicing and even DNA repair [1]. Argonautes (Agos) are also present in many bacterial and archaeal species, and previous genomic studies showed that the diversity of prokaryotic Agos (pAgos) is far greater than that of eAgos [2,3]. pAgos are one of the tools for cells to defend against virus infection or foreign DNA invasion [2,3,4]. Furthermore, pAgos can be divided into three clades, including long pAgos, short pAgos, and PIWI-RE proteins [5]. The conserved core structure of most long pAgos includes the *N*-terminal, L1, PAZ, L2, MID, and PIWI domains, while short pAgos consist of only MID and PIWI domains [5]. Active pAgos have a complete catalytic tetrad DEDX (X is D, H, or N) in the PIWI domain, which binds divalent metal ions and participates in catalysis [6,7,8]. When programmed with small nucleic acid guides, active pAgos can cleave the complementary target at a site between the position of 10 and 11 nucleotides starting from the 5′-end of the guide [9].

Given that the nuclease cleavage activities of CRISPR-Cas (clustered regularly interspaced short palindromic repeats-CRISPR associated) proteins have changed the genome-editing field, pAgos with their powerful capacities may as well become valuable tools for applications in biotechnology [10,11,12,13]. Furthermore, pAgos have remarkable advantages when compared with CRISPR-Cas nucleases, the guide-directed recognition cleavage of pAgos is mediated by guide-target pairing without the restriction of PAM motifs, and the synthesis of short DNA and RNA guides utilized by pAgos is relatively cheap [10,12,13]. Therefore, theoretically we can design high-throughput guides to specifically target any desired location. To date, only a few pAgos derived from mesophilic bacteria have been reported to perform single-stranded DNA (ssDNA) and double-stranded DNA (dsDNA) cleavage at 37 °C, including CbAgo, LrAgo, SeAgo, CpAgo, IbAgo and KmAgo [14,15,16,17,18]. However, these pAgos exhibit low levels of nuclease activity on dsDNA, and preferentially cleave negatively supercoiled plasmids and/or DNA fragments with low GC-content [14,19]. Among them, only KmAgo from mesophilic bacteria can efficiently cleave single-stranded RNA, and in view of the SARS-CoV-2 pandemic [20], it is mandatory to further explore RNA cleavage. Meanwhile, considering that pAgos have been implicated in the cell’s defense against mobile genetic elements [9,21] and pAgos can cleave the single-stranded nucleic acid targets and dsDNA in vitro, they have the powerful potential for genome manipulation.

Considering the potential application of mesophilic pAgos in genome editing and RNA manipulation, we proceed here to identify mesophilic pAgos with desired biochemical cleavage function and high specificity. In this work, we characterize three pAgos with about 50% sequence identity with KmAgo, and RsuAgo from *Rummeliibacillus suwonensis* is mainly explored owing to the ability to recognize both DNA and RNA targets. In addition, RsuAgo-mediated RNA target cleavage occurs with only a short guide (16–20 nt), and RsuAgo-mediated RNA cleavage is extremely sensitive to single nt mismatches in the guide-target duplex. Furthermore, we demonstrate that RsuAgo can perform precise guide-dependent cleavage of up to about 45% of superhelical dsDNA at 37 °C. The programmable DNA and RNA nuclease activity of RsuAgo broadens our understanding of pAgos and can promote the development of pAgos-based applications at moderate temperatures.

## 2. Materials and Methods

### 2.1. Protein Expression and Purification

DNA sequences encoding RsuAgo (WP_146547607.1), BsAgo (WP_100402176.1), DeAgo (WP_052698403.1), and PmAgo (WP_068488195.1) were codon-optimized for expression in *E. coli* and cloned into pET28a expression vector containing an *N*-terminal 6 × His tag by Wuhan Genecreate Biotechnology Co. Ltd., Wuhan, China. DNA sequence encoding BsAgo was cloned into a custom pET-based expression vector containing an *N*-terminal 6 × His tag, GST tag and HRV-3C protease cleavage site and verified by DNA sequencing (Appendix A). Mutations were introduced into RsuAgo to obtain RsuAgo_DM (RsuAgo double mutant; D530A, D599A) using PCR-mediated site-directed mutagenesis and verified by DNA sequencing. Proteins were expressed in *E. coli* BL21 (DE3) (Novagen, Beijing, China). For protein expression, cells were cultured in LB (Luria-Bertani) medium containing 50 μg/mL kanamycin to an OD_600_ of 0.6, expression was induced by addition of IPTG (isopropyl-β-D-1-thiogalactopyranoside) to 0.5 mM final concentration, and cells were incubated at 18 °C while shaking for 20 h. The cells were collected by centrifugation, washed with wash buffer (500 mM Tris-HCl pH 7.5, 0.5 M NaCl), and stored at −80 °C for further protein purification.

The cells were lysed by sonication (SCIENTZ-IID, 400 W, 2 s on/4 s off for 15 min) in Buffer A (20 mM Tris-HCl, pH 7.5, 500 mM NaCl, and 10 mM imidazole), and then the supernatant was collected by centrifugation at 14,000 rpm for 30 min and then the supernatant was bound to Ni-NTA agarose which had been activated by 10 mM imidazole. The resin was washed extensively with Buffer A and Buffer A containing 50 mM imidazole, and bound protein was eluted with Buffer A containing 300 mM imidazole. The eluted protein was concentrated against Buffer B [20 mM HEPES (N-2-Hydroxyethylpiperazine-N-2-Ethane Sulfonic Acid) pH 7.5, 500 mM NaCl and 1 mM DTT (dithiothreitol)] by ultrafiltration using an Amicon 50K filter unit (Millipore, Boston, MA, USA). 

For BsAgo, 6 × His tag and GST tag was removed by cleavage with HRV 3C overnight at 4 °C and the cleaved proteins were separated from the fusion tag using a second Ni-NTA affinity step. For DeAgo and PmAgo, proteins were diluted in 20 mM HEPES pH 7.5 to lower the final salt concentration to 125 mM NaCl. The diluted proteins were applied to a Heparin column (HiTrap Heparin HP, GE Healthcare, Boston, MA, USA) equilibrated with Buffer C (20 mM HEPES pH 7.5, 125 mM NaCl and 1 mM DTT), washed with at least 10 column volumes of the same buffer and eluted with a linear NaCl gradient (0.125–2 M). Fractions containing pAgos were concentrated by ultrafiltration using an Amicon 50K filter unit (Millipore, Boston, MA, USA). For RsuAgo, these fractions were further loaded onto a size exclusion column (Superdex 75 16/600 column, GE Healthcare, Boston, MA, USA) and eluted with Buffer D (20 mM HEPES, pH 7.5, 500 mM NaCl, and 1 mM DTT). Proteins were detected by SDS-PAGE and the collected proteins were concentrated by Amicon 50K filter unit (Millipore, Boston, MA, USA). Protein concentration was assessed by Thermo Scientific NanoDrop 8000 Spectrophotometer. Purified proteins were aliquoted and frozen in liquid nitrogen and stored in an ultra-low temperature refrigerator at −80 °C.

### 2.2. Preparation of 5′-Phosphorylated DNA and RNA Guides

Synthetic oligonucleotide guides were diluted to 100 pmol/uL in DEPC (diethylpyrocarbonate)-treated water and 5′-phosphorylated with T4 PNK (T4 polynucleotide kinase), the 5′-phosphorylated ssDNA guide, and single-stranded RNA (ssRNA) guide were referred to as P-gDNA and P-gRNA, respectively, while the unmodified ssDNA and ssRNA was referred to as OH-gDNA and OH-gRNA, respectively. For the study of cleavage activity affected by divalent metal ions, P-gDNA and P-gRNA were synthesized directly to avoid the potential interference of metal ions in the T4 PNK phosphorylation reaction buffer. 

### 2.3. Phylogenetic Tree and Sequence Alignment of RsuAgo

We chose the protein sequence of KmAgo (WP_010289662.1) as the query and used the web-based BLASTp algorithm of the NCBI database to search for pAgos with relatively high sequence identity, and pAgo sequences were aligned for phylogenetic analysis. The evolutionary relationship between the four pAgos and several other characterized pAgos was analyzed with the MEGA 7.0 software [22], (Temple University, Philadelphia, PA, USA).

### 2.4. Single-Stranded Nucleic Acid Cleavage Assay

The single-stranded nucleic acid cleavage experiments were performed at 37 °C with the molar ratio of pAgo: guide: target of 4:2:1. 800 nM pAgos and 400 nM guides were mixed in the reaction buffer containing 10 mM HEPES-NaOH pH 7.5, 100 mM NaCl, 5 mM MnCl_2_, 5% glycerol and incubated at 37 °C for 10 min to form the Ago-guide complex, and then the 200 nM target were added. After reaction for the indicated time interval, the reaction was stopped by mixing the sample with an equal volume of a termination solution (95% formamide, 18 mM EDTA, 0.025% sodium dodecyl sulfate, and 0.025% bromophenol blue) and heating it at 95 °C for 5 min. The cleavage products were separated by 20% denaturing polyacrylamide gel electrophoresis (PAGE) and stained with SYBR Gold (Invitrogen, Carlsbad, CA, USA). Kinetic analysis of target cleavage was performed under single-turnover conditions using target substrates containing a fluorescent dye at the 5′-end (FAM, for analysis of fully complementary 18 nt guide), and the kinetics of cleavage were monitored over time. Thereafter, the denaturing PAGE gels were directly visualized with gel Doc^TM^ XR+ (Bio-Rad, Hercules, CA, USA) without staining and then analyzed with ImageJ and Origin software.

To test whether the four pAgos act uniformly on DNA and RNA targets mediated by guides in vitro, we used a set of synthetic guides and targets to analyze cleavage activity. gDNA or gRNA was loaded onto pAgos, and then Ago-guide complex was incubated with DNA target (tDNA) or RNA target (tRNA). Subsequently, we mainly carried out a series of explorations on the cleavage activity of RsuAgo. To investigate whether the 5′-end nucleotide of the guide affects the activity of RsuAgo, substrate cleavage kinetics were tested by providing guides containing a 5′-A, 5′-C, 5′-G, or 5′-T (5′-U) residue at the 5′-end. The effects of divalent metal ions on RsuAgo activity were assessed in the presence of different ions (Mg^2+^, Ca^2+^, Mn^2+^, Co^2+^, Ni^2+^, Cu^2+^, Zn^2+^, EDTA). To investigate the optimum cleavage concentration of divalent metal ions, we performed gradient experiments of reaction buffer with a final Mg^2+^ and Mn^2+^ concentration of 0 mM, 0.05 mM, 0.1 mM, 0.5 mM, 1 mM, 2.5 mM, 5 mM, and 10 mM. To investigate the effects of guide length on cleavage activity, we tested a series of gDNAs from 9 to 40 nt length. To analyze the temperature dependence of target cleavage, pAgos were combined with 5′-T gDNA (T-gDNA) or 5′-A gDNA (A-gDNA) and then used to cleave 5′-T tDNA (T-tDNA) or 5′-A tRNA (A-tRNA), respectively. Then all samples were incubated simultaneously at indicated temperatures using the Polymerase Chain Reaction Thermal Cycle Instrument (T100, Bio-Rad). To study the effects of guide-target duplex mismatches, we designed a set of gDNAs with single nucleotide mismatches in the 5′-A-gDNA from positions 1–18, respectively (Appendix A). All cleavage experiments were carried out in triplicates.

### 2.5. Double-Stranded DNA Cleavage with RsuAgo

In two half-reactions, 8 pmol of RsuAgo was loaded with either 10 pmol of forward or reverse P-gDNA in reaction buffer (5 mM HEPES-NaOH pH 7.5, 0.5 mM MnCl_2_, 2.5% glycerol) supplemented with 5 mM NaCl if not stated otherwise. The two reaction mixtures were incubated for 30 min at 37 °C. Next, both half-reactions were mixed and 200 ng of plasmid DNA were added, after that the mixture was incubated at 37 °C for 2 h. Samples were mixed with 6 × loading buffer (NEB) before being resolved on 1% agarose gels. All nucleic acids used in this study are listed in Appendix A. When exploring cleavage of plasmid DNA as a function of NaCl concentration, considering that the storage buffer of RsuAgo protein contained 500 mM NaCl, we prepared reaction buffer containing 1000 mM, 500 mM, 250 mM, 100 mM NaCl, and thus the final NaCl concentration of the reaction buffer was 150 mM, 100 mM, 75 mM, 60 mM, 55 mM, respectively. 

## 3. Results and Discussion

### 3.1. Phylogenetic Tree and Sequence Alignment of RsuAgo

The search for mesophilic pAgos was performed using the web interface of the BLASTp program, with the protein sequence of KmAgo (WP 010289662.1) as a query. RsuAgo, BsAgo, DeAgo, and PmAgo were chosen as candidates for their high sequence identity (about 50%) and aligned with KmAgo. Phylogenetic analysis revealed that RsuAgo, BsAgo, PmAgo, and DeAgo are closely related to KmAgo, while PmAgo is more distantly related to the former three pAgos (Figure 1A). Multiple sequence alignment showed that RsuAgo, BsAgo, DeAgo, and PmAgo contain a conserved catalytic DEDX tetrad in the PIWI domain, indicating that they might have catalytic activity (Figure 1B).

### 3.2. Single-Stranded Nucleic Acid Cleavage Assay

RsuAgo, RsuAgo_DM, BsAgo, DeAgo, and PmAgo were expressed in *E. coli* BL21 (DE3) and analyzed by SDS-PAGE. The purified pAgo proteins were consistent with the predicted molecular weights (Appendix A). Then, to confirm whether RsuAgo, BsAgo, DeAgo, and PmAgo are indeed active nucleases, we performed in vitro cleavage assays in which pAgos were loaded with gDNA or gRNA. A complementary 45 nt tDNA or tRNA was added, followed by incubation at 37 °C in a reaction buffer containing 5 mM Mn^2+^. BsAgo and DeAgo did not show RNA cleavage activity and only showed DNA cleavage activity guided by P-gDNA (Appendix A). For PmAgo, no cleavage activity at all was observed (Appendix A). RsuAgo was not only able to cleave both DNA and RNA guided by P-gDNA, but also cleaved DNA guided by OH-gDNA and P-gRNA (Figure 2A). 

Considering that the canonical cleavage site of pAgos is located between the tenth (10′) and eleventh (11′) nucleotides starting from the guide 5′-end [7], we synthesized 34 nt long FAM-labeled ssDNA or ssRNA as markers, and their sequences corresponded to the predicted cleavage products (Figure 2B, Appendix A). We observed target cleavage by BsAgo, DeAgo, and RsuAgo at a single site between target positions 10’ and 11’, resulting in the 34 nt long 5′-fragment of the target (Figure 2A and Appendix A). Previous studies showed that active pAgos were dependent on the complete catalytic DEDX tetrad in the PIWI domain [7,8,23], and double mutations in the tetrad eliminated the cleavage activity of RsuAgo (Appendix A). In this and subsequent experiments, we mainly explored RsuAgo due to its remarkable DNA and RNA cleavage activity (Figure 2A), which could be potentially used to manipulate both DNA and RNA. Furthermore, it is important to highlight here the RsuAgo-mediated DNA cleavage guided by P-gRNA, because this type of cleavage has been rarely reported except for KmAgo, especially for mesophilic pAgos [17,18,24]. 

### 3.3. Kinetic Analysis of Single-Starnded Nucleic Acid Cleavage by RsuAgo

To study RsuAgo-mediated target cleavage by different types of guides, we performed cleavage kinetic experiments concerning the fundamental properties of P-gDNA: tRNA, P-gDNA: tDNA, P-gRNA: tDNA, and OH-gDNA: tDNA with 5 mM Mn^2+^ at 37 °C in single-turnover conditions (RsuAgo-guide complex in excess over tDNA). We found that the fastest cleavage rate was observed for DNA cleavage guided by P-gDNA, followed by DNA cleavage guided by OH-gDNA and RNA cleavage guided by P-gDNA (Figure 3A). The cleavage kinetic analysis verified that gDNA-mediated DNA cleavage started with a burst phase and then entered the platform phase. Interestingly, while in other cases (P-gDNA: tRNA, P-gRNA: tDNA), a slower and rate-limiting phase was observed instead of a burst phase at the beginning, reflecting the slow formation of the catalytically competent complex [14]. Together, as long as the guide or target contained RNA, almost no cleavage activity was observed in the first five minutes. The reaction rate of DNA cleavage guided with OH-gDNA was only slightly lower in comparison to P-gDNA and higher in comparison to P-gRNA. The cleavage rate of RsuAgo using the P-gRNA guides was the lowest. Therefore, RsuAgo prefers to use P-gDNA as a guide strand. The initial burst phase reflects the quick rate of the ternary complex formation and catalysis, and the absence of the initial burst phase may reflect a slower rate of the ternary complex formation and/or catalysis by RsuAgo [14,25,26,27].

### 3.4. Effects of 5′-Nucleotide of Guide on RsuAgo Cleavage Activity

Previous pAgo studies reported that a particular nucleotide residue at the first position of the guide, also known as an anchoring area, was accommodated in a protein pocket formed by the MID domain [4,18,28,29]. In addition, certain eAgos and pAgos showed a bias for a specific guide 5′-nucleotide [19]. To determine whether RsuAgo had a preference for the 5′-nucleotide of gDNA or gRNA, we tested several different 5′-end nucleotides of guide variants but otherwise identical sequences. According to time course analysis of tDNA cleavage, a purine (A or G) was preferred over pyrimidines, and a 5′-C resulted in the slowest kinetics (Figure 3B and Appendix A). The preference is similar to MjAgo, which prefers a 5′-purine base of the gDNA (i.e., 5′-A and 5′-G gDNA) [28]. Interestingly, the kinetics of P-gDNA-mediated tRNA cleavage was the fastest with 5′-A and the slowest with 5′-G (Figure 3C and Appendix A), similarly, 5′-A P-gRNA-mediated tDNA cleavage was the fastest (Figure 3D and Appendix A). Despite these preferences, in vitro cleavage assays showed that P-gDNA or P-gRNA containing a 5′-C, 5′-T, or 5′-G nucleotide resulted in substantial tDNA cleavage activity (Figure 3B,D). In conclusion, RsuAgo prefers P-gRNA or P-gDNA with 5′-A, and the preference mechanism for the 5′-A needs further exploration based on structure analysis. Despite this preference, RsuAgo-mediated target cleavage still maintains activity guided by other different types of 5′-end nucleotide guide, indicating that RsuAgo could be easily used to cleave DNA or RNA without obvious sequence restrictions, thus this work paves the way for further development of DNA and RNA manipulation tools. 

In the initial experiment, we performed a 1 h cleavage reaction and P-gRNA with 5′-U was used, and DNA cleavage products were barely visible (Appendix A). Later, we found that gDNA with different 5′-end nucleotides affected their cleavage activity when tRNA was cleaved. Inspired by the 5′-end nucleotide preference we redesigned gRNA with different 5′-end nucleotides. Surprisingly, gRNA-programmed RsuAgo could perform DNA cleavage when the reaction time was prolonged, but its cleavage activity was poor (Figure 3D). As expected, gRNA-mediated cleavage efficiency was significantly lower than that mediated by gDNA. Moreover, there is only a weak base preference for the 5′-terminal gRNA nucleotide in gRNA-mediated tDNA cleavage. 

### 3.5. Effects of Different Divalent Metal Ions on RsuAgo Cleavage Activity

Considering that divalent metal ions are important cofactors for pAgo activity, we investigated the effects of divalent metal cations on the kinetics of guide-dependent tDNA or tRNA cleavage for 1 h at 37 °C. For tDNA cleavage, OH-gDNA mediated cleavage with decreasing efficiency at ≤1 mM Mn^2+^, whereas cleavage with P-gDNA was quite constant from 0.05 to 10 mM Mn^2+^ (Appendix A). Titration of Mg^2+^ ions showed no cleavage activity with OH-gDNA at ≤1 mM Mg^2+^, while RsuAgo could cleave with P-gDNA from 0.1 to 10 mM Mg^2+^ (Appendix A). In contrast to pAgo-mediated RNA cleavage by KmAgo [17], RsuAgo was unable to use OH-gDNA as a guide for RNA cleavage. In addition, P-gDNA-mediated RNA cleavage experiments revealed differences in the utilization of Mn^2+^ and Mg^2+^ ions. RsuAgo cleaved RNA only at Mg^2+^ ions concentrations above 1 mM (Appendix A), while it could perform RNA cleavage at Mn^2+^ ions concentrations ranging from 0.05 mM to 10 mM, with comparable efficiency (Appendix A).

For tDNA cleavage, RsuAgo guided with P-gDNA was active with both Mn^2+^ and Mg^2+^ (Figure 4A). For tRNA cleavage, RsuAgo guided with P-gDNA had the highest endonuclease activity in the presence of Mn^2+^, but was much less active with Mg^2+^ (Figure 4B). Furthermore, no cleavage activity was observed for both substrates, tDNA and tRNA, in the presence of Ca^2+^, Co^2+^, Ni^2+^, Cu^2+^ or Zn^2+^, and EDTA for both cleavage conditions of tDNA or tRNA. We further compared the kinetic cleavage efficiency affected by Mg^2+^ and Mn^2+^ when guides were P-gDNAs. For tDNA cleavage, an increase phase was detected under Mn^2+^ conditions during the first minute (Figure 4C), interestingly, a prolonged lag phase (>5 min) was observed followed by a slower increase phase under Mg^2+^ conditions. Thus, RsuAgo-mediated cleavage in the Mn^2+^ buffer was more efficient than in the Mg^2+^ buffer. However, for tRNA cleavage, we observed that the Mn^2+^-assisted cleavage reaction started with a lag phase (Figure 4D) and this lag phase was extended from ~10 to 30 min in Mg^2+^ buffer. We propose that the lag phase reflects slow build-up of the ternary and/or catalytically active complex in RsuAgo-mediated RNA cleavage [17]. In conclusion, the cleavage efficiency mediated by Mn^2+^ was better than that mediated by Mg^2+^.

### 3.6. Effects of Temperature and Guide Length on RsuAgo Activity

Analysis of tDNA cleavage activity at different temperatures showed that RsuAgo had a comparable level of nuclease cleavage between 25 °C and 60 °C and still maintained a certain activity at 65 °C (Figure 5A). RsuAgo-mediated RNA cleavage was significantly less effective than DNA cleavage at most temperatures, and the RNA target and product bands were rather faint (Figure 5B). We speculate that the degradation was caused by Mn^2+^, since the RNA target and product signals decreased with increasing Mn^2+^ concentration (Appendix A). Considering the practical application of RNA manipulation in the future, we can reduce the Mn^2+^ concentration to avoid the degradation of the target and product bands in the later research, and we need to further explore other pathways to prevent degradation of RNA targets. In conclusion, RsuAgo was active in DNA and RNA cleavage at moderate temperatures and also performed cleavage at elevated temperatures.

Then we investigated the role of guide length by testing a series of P-gDNAs from 9 to 40 nt that shared identical sequences at their 5′-ends so that the predicted cleavage site was the same for all guides. We found that 15–25 nt long P-gDNAs were sufficient for RsuAgo-mediated DNA target cleavage (Figure 5C), which is similar to that of CbAgo [30]. Previous structural and biochemical studies on CbAgo showed that it can bind a single-stranded guide with a minimum length of 12 nt, but it could only cleave tDNA in the presence of gDNA with a minimum length of 15 nt [30,31]. Interestingly, in contrast to previously reported pAgos that perform RNA cleavage over a wide range of 9–40 nt P-gDNA, such as KmAgo [17], RsuAgo cleaved RNA only with short 16–20 nt P-gDNAs (Figure 5D), and a more extended guide length essentially abolished cleavage activity. It is known that FpAgo promotes DNA cleavage only with short 15–20 nt gDNAs, rather than tolerating a wide range of guide lengths [32], but restrictions in the guide length for RNA cleavage by pAgos have not yet been reported.

### 3.7. Effects of Mismatches in the Guide-Target Duplex on the Cleavage Activity of RsuAgo

Previous studies of eAgos and several mesophilic pAgos showed that Agos identify their target via a guide oligonucleotide that forms Watson-Crick base pairs with the target, and mismatches between guide and target strands might have large effects on cleavage efficiency and precision [14,31,33,34]. Previous studies of mesophilic pAgos, including CbAgo and KmAgo, showed that mismatches between the guide and target nucleic acids might have effects on target cleavage [14,17,18,30]. Here we explored the mismatches on cleavage by introducing single nt mismatches at positions from nucleotide 1 (m1) to nucleotide 18 (m18) in the 18 nt guide strand (Figure 6A). For tDNA cleavage, single nt mismatches at m3 and m12 significantly reduced DNA cleavage efficiency of RsuAgo, while single nt mismatches were fully tolerated in the other regions of gDNA (Figure 6B and Appendix A). 

Interestingly, we observed that weaker pairing strength in its seed region (m2, m3) did not impair RsuAgo-mediated cleavage activity, but rather stimulated tRNA cleavage (Figure 6C,D and Appendix A). Moreover, RsuAgo-mediated tRNA cleavage was extremely sensitive to single nt mismatches in the seed region (m4, m5, m7, m8), central region (m9, m11, m12) and supplementary region (m14, m15). Thus, mutation at most positions dramatically reduced or nearly abolished RNA cleavage activity, with the exception of 6 positions (m6, m10, m13 and m16-m18 in the 3′-tail region [18]). (Figure 6C,D and Appendix A). To the best of our knowledge, in comparison, mismatches at m7, m8, and m14 significantly reduced target RNA cleavage and those at m9-m13 almost eliminated RNA cleavage activity catalyzed by KpAgo from the yeast *Kluyveromyces polysporus* [33]. In the case of KmAgo, dinucleotide mismatches m11/m12 (tDNA cleavage) and m8/m9 (tRNA cleavage) caused the strongest decreases in cleavage efficiency [17].

Here we have characterized, for the first time, RsuAgo possessed gDNA/tDNA mismatch tolerance but was sensitive to mismatches at only 2 sites, and catalyzed DNA-guided RNA cleavage activity that was extremely sensitive to mismatches at 9 sites of the gDNA. We further showed that introduction of single nt mismatches (m2, m3) into the seed region enhances RNA cleavage and mismatches at 9 other positions eliminate RNA cleavage activity, which provides important information for applications of RsuAgo in RNA manipulation. likewise, for mouse Ago2 intentionally weakened seed pairing enhanced cleavage activity relative to the guide-target duplex with full complementarity [25,35]. 

### 3.8. RsuAgo Mediates Cleavage of Double-Stranded DNA

Previous research showed that mesophilic pAgos could cleave negatively supercoiled plasmid but not the linearized plasmid, which suggested that the cleavage relyed on the negatively supercoiled state of a plasmid to facilitate local DNA melting before cleavage could take place [17,18]. Studies to date have used supercoiled plasmid DNA as a model template for dsDNA cleavage and GC-content of the target DNA plays a role during DNA targeting by mesophilic pAgos [16]. Low GC-content regions of plasmids could be moderately cleaved at 37 °C, while high GC-content regions of plasmids could not be cleaved, possibly because AT-rich DNA is more prone to unwinding. It is speculated that higher temperatures promote increased DNA strand separation [19,36] such that pAgos can effectively create “double-strand breaks” in dsDNA by targeting and cleaving complementary DNA strands using two separate guide/pAgo complexes.

When we started to explore experiments of RsuAgo-mediated dsDNA plasmid cleavage with reaction buffer containing 100 mM NaCl, we found that the plasmid cleavage activity was weak. Considering that the plasmid cleavage activity is affected by NaCl concentration [36], we then explored the effects of NaCl concentration on RsuAgo-mediated dsDNA plasmid cleavage. As expected, RsuAgo-mediated dsDNA plasmid cleavage was sensitive to NaCl, with 55–100 mM as the optimal concentration, whereas a higher concentration of NaCl reduced cleavage efficiency (Figure 7A), indicating that NaCl concentration plays an important role in the plasmid cleavage activity of RsuAgo. Furthermore, it has been confirmed that at high NaCl concentrations, positively charged sodium ions mask the negative charge of the DNA, reducing the repulsion (‘‘breathing’’) between the two DNA backbones [37,38].

To further explore whether the GC-content of the dsDNA plasmid plays a role during dsDNA cleavage by RsuAgo, we designed a series of guides targeting different GC-content regions of the pUC19 plasmid (Appendix A). When RsuAgo was preincubated with a gDNA, RsuAgo nicked the plasmid DNA (Figure 7B). Only when supplied with a pair of gDNAs, the linear plasmid cleavage product appeared. Furthermore, RsuAgo-gDNA complexes were able to efficiently generate dsDNA breaks in regions with a GC-content of 45% or lower (Figure 7B), but not (or barely) regions with higher GC-content (Appendix A). These results demonstrate that RsuAgo is able to mediate DNA-guided cleavage of plasmid dsDNA, the efficiency depending on NaCl and GC-content of the dsDNA target.

## 4. Conclusions

As kmAgo was reported to be programmable omnipotent mesophilic pAgo, this work demonstrated that three mesophilic pAgos with about 50% sequence identity to KmAgo possessed typical gDNA-mediated DNA cleavage activity. Among them, RsuAgo could not only cleave DNA in a P-gDNA-dependent sequence-specific manner but also cleave RNA under the guidance of P-gDNA. Furthermore, there was a weak DNA cleavage activity guided by P-gRNA. In addition, RsuAgo preferred a guide with a 5′-A residue, but this bias was moderate, since guides with a 5′-C, 5′-T or 5′-G residue gave also rise to substantial target cleavage. RsuAgo could cleave DNA with gDNAs in the length range of 15–25 nt. However, in the case of RNA cleavage, the length range of gDNAs mediating RNA cleavage was confined to 16–20 nt. Moreover, RsuAgo showed a strong mismatch sensitivity in gDNA-mediated RNA cleavage. Mismatches at 9 single positions of the guide-target duplex strongly reduced or even eliminated cleavage by RsuAgo. In addition, RsuAgo could effectively mediate gDNA-directed dsDNA plasmid linearization at 37 °C and was able to cleave dsDNA target sites with up to 45% GC-content. These findings broaden our understanding of mesophilic pAgos and pave the way for application in RNA and DNA.

## Figures and Tables

**Figure 1 biomolecules-12-00355-f001:**
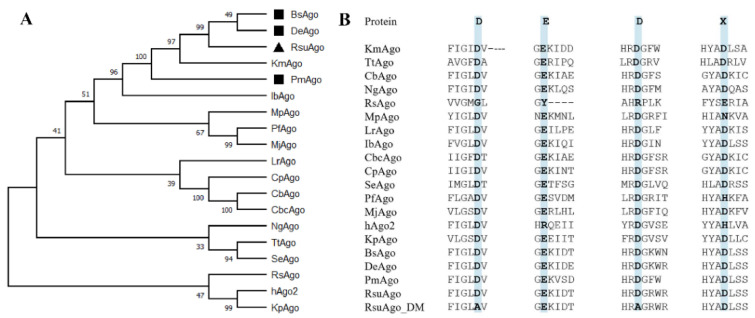
Sequence analysis and phylogenetic tree visualizations. (**A**) Maximum-likelihood phylogenetic tree analysis of RsuAgo based on amino acid sequences. The numbers at the nodes indicate the bootstrap values for the maximum likelihood analysis of 1000 resampled data sets. RsuAgo is indicated as a black triangle, BsAgo, DeAgo, and PmAgo are indicated as a black rectangle. (**B**) Multiple sequence alignment of RsuAgo, DeAgo, BsAgo, and PmAgo with several other characterized pAgo proteins. The catalytically dead variant of the RsuAgo protein (RsuAgo_DM) with amino acid substitutions in the catalytic tetrad is shown.

**Figure 2 biomolecules-12-00355-f002:**
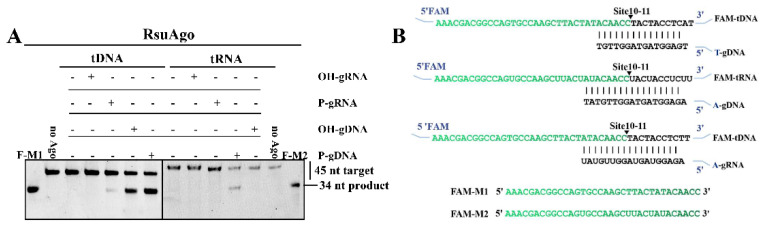
Analysis of single-stranded nucleic acid cleavage activity of pAgos. (**A**) Cleavage by RsuAgo with FAM-labeled DNA and RNA targets. (**B**) The sequence of the synthetic miRNA-based guide and target sequences that were used for the in vitro cleavage assays. The black triangle indicates the cleavage site. The vertical lines indicate contiguous Watson-Crick pairing. F-M1 (FAM-M1) and F-M2 (FAM-M2) are chemically synthesized, 34 nt long ssDNA (F-M1) and ssRNA (F-M2) oligonucleotides with a FAM-label at the 5′-end that were loaded on gels as size markers corresponding to the length of the DNA and RNA cleavage products, respectively. All experiments were performed at the 4:2:1 RsuAgo: guide: target molar ratio in reaction buffer containing Mn^2+^ ions for 1 h at 37 °C. All cleavage experiments were carried out in triplicates.

**Figure 3 biomolecules-12-00355-f003:**
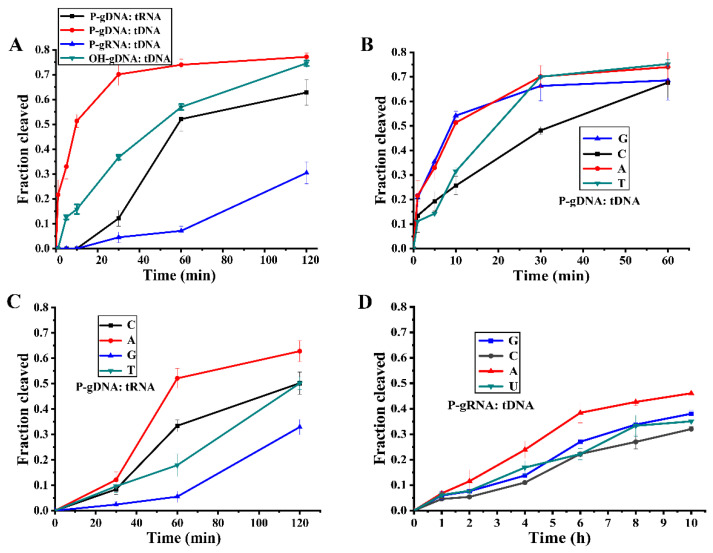
Effects of gDNA modification and 5′-end nucleotide on RsuAgo cleavage activity. (**A**) Cleavage kinetics on DNA and RNA targets using DNA-guided and RNA-guided RsuAgo. (**B**) Cleavage kinetics on DNA target using different 5′-end nucleotides of DNA guide. (**C**) Cleavage kinetics on RNA target using different 5′-end nucleotides of DNA guide. (**D**) Cleavage kinetics on DNA target using different 5′-end nucleotides of RNA guide. All experiments were performed at the 4:2:1 RsuAgo: guide: target molar ratio in reaction buffer containing 5 mM Mn^2+^ ions at 37 °C. All cleavage experiments were carried out in triplicates and quantified. Error bars represent the SD based on three independent experiments.

**Figure 4 biomolecules-12-00355-f004:**
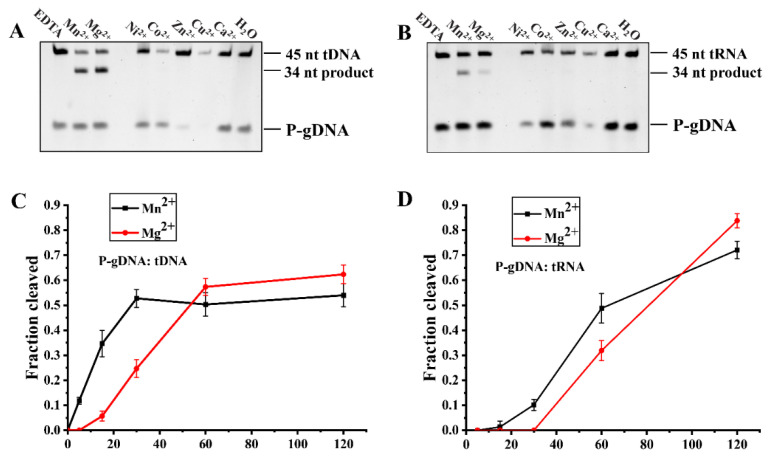
Effects of different divalent metal ions on RsuAgo cleavage activity. (**A**) Effects of different divalent metal cations on DNA target cleavage activity mediated by P-gDNA. (**B**) Effects of different divalent metal cations on RNA target cleavage activity mediated by P-gDNA. (**C**) Cleavage kinetics efficiency of DNA target mediated by Mn^2+^ and Mg^2+^. (**D**) Cleavage kinetics efficiency of RNA target mediated by Mn^2+^ and Mg^2+^. All experiments were performed at the 4:2:1 RsuAgo: guide: target molar ratio in reaction buffer containing different divalent metal ions for 1 h at 37 °C. All cleavage experiments were carried out in triplicates and quantified. Error bars represent the SD based on three independent experiments.

**Figure 5 biomolecules-12-00355-f005:**
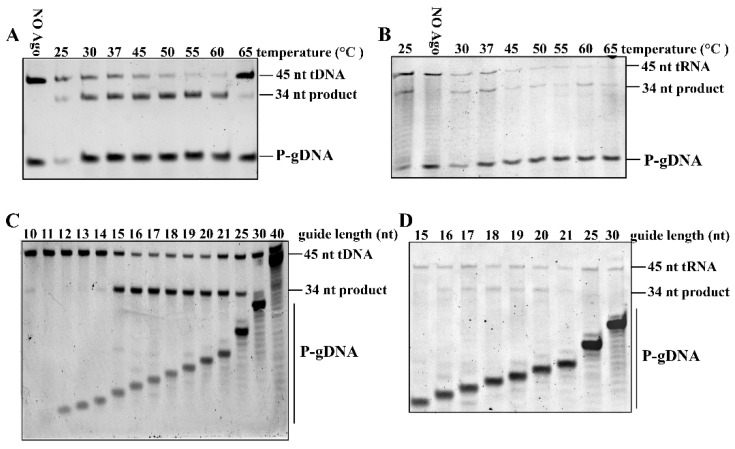
Effects of temperature and guide length on RsuAgo activity. (**A**) Effects of temperature on DNA target cleavage guided by P-gDNA. (**B**) Effects of temperature on RNA target cleavage guided by P-gDNA. (**C**) Effects of P-gDNA length on DNA cleavage activity. (**D**) Effects of P-gDNA length on RNA cleavage activity. All experiments were performed at the 4:2:1 RsuAgo: guide: target molar ratio in reaction buffer containing 5 mM Mn^2+^ ions for 1 h at 37 °C. All cleavage experiments were carried out in triplicates.

**Figure 6 biomolecules-12-00355-f006:**
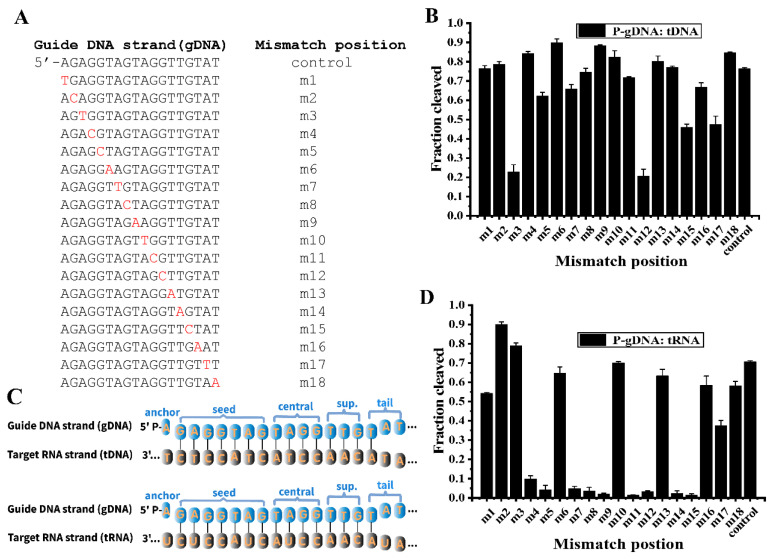
Effects of mismatches in the guide-target duplex on the cleavage activity of RsuAgo. (**A**) Schematic diagram of guide single nucleotide mismatch sites, mismatched positions are indicated in red. (**B**) Effects of single nucleotide mismatches in the P-gDNA: tDNA duplex on the slicing activity of RsuAgo. (**C**) Sequences of the gDNA (blue) and DNA/RNA target (grey). The different functional gDNA elements are indicated above the gDNA; sup., supplementary region. (**D**) Effects of single nucleotide mismatches in the P-gDNA: tRNA duplex on the slicing activity of RsuAgo. Control, gDNA with full complementarity to the target. All experiments were performed at the 4:2:1 RsuAgo: guide: target molar ratio in reaction buffer containing 5 mM Mn^2+^ ions for 1 h at 37 °C. All cleavage experiments were carried out in triplicates and quantified. Error bars represent the SD based on three independent experiments.

**Figure 7 biomolecules-12-00355-f007:**
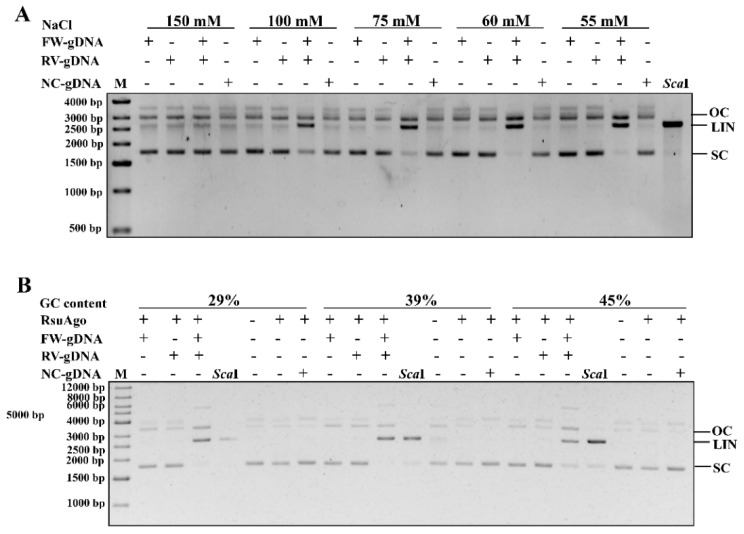
Cleavage of double-stranded plasmid DNA by RsuAgo. (**A**) RsuAgo-mediated cleavage of plasmid pUC19 at different NaCl concentrations. RsuAgo and a gDNA with 29% GC-content were preincubated for 30 min at 37 °C, followed by addition of pUC19 dsDNA, incubation for 2 h at 37 °C in reaction buffer containing 0.5 mM Mn^2+^ and cleavage analysis by 1% agarose gel electrophoresis. (**B**) RsuAgo-mediated plasmid pUC19 cleavage with gDNAs differing in their GC-content in reaction buffer containing 0.5 mM Mn^2+^ and 55 mM NaCl for 2 h at 37 °C. *Sca*I lane: the isolated plasmid was digested with *Sca*I for 2 h. FW/RV-gDNA, forward and reverse gDNA corresponding to a specific target site in plasmid pUC19; NC-gDNA, a non-complementary gDNA control; M, molecular weight marker; LIN, linearized plasmid; SC, supercoiled plasmid; OC, open circular plasmid. All cleavage experiments were carried out in triplicates.

## Data Availability

Not applicable.

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
