# Peer review of "Characterization of a Programmable Argonaute Nuclease from the Mesophilic Bacterium Rummeliibacillus suwonensis"

_biomolecules, 2022, doi:10.3390/biom12030355_

Round 1
Reviewer 1 Report
Title: “Characterization of a programmable Argonaute nuclease from the mesophilic bacterium Rummeliibacillus suwonensis”
In this article, the authors have reported that three active pAgos, which have about 50% sequence identity with KmAgo, possess typical DNA-guided DNA target cleavage activity. Among them, RsuAgo from Rummeliibacillus suwonensis is mainly explored for which can cleave both DNA and RNA targets. RsuAgo-mediated RNA target cleavage adopts with only a short guide 16-20 nt length rather than a broad guide selectivity, and mismatches between the guide and target sequence greatly affect the efficiency of RNA target cleavage. Also, authors have identified RsuAgo can cleave the double stranded DNA at low concentrations of NaCl at 37 °C. These properties of RsuAgo provide a new tool for DNA and RNA manipulation at moderate temperatures. To further explore whether the GC-content of the dsDNA plasmid plays a role during dsDNA cleavage by RsuAgo, authors have designed a series of guides targeting different GC-content regions of the pUC19 plasmid. When RsuAgo was preincubated with a gDNA, RsuAgo nicked the plasmid DNA. Also, RsuAgo-gDNA complexes were able to efficiently generate dsDNA breaks in regions with a GC-content of 45% or lower (Figure 6B), but not regions with higher GC-content. I suggest the authors to improve the quality of the figures throughout the manuscript. The authors have written this manuscript very well and based on the importance of this work it could be published. I suggest publishing in Bimolecules.
Author Response
Thank you very much for your kind comments on our manuscript. These is no doubt that these comments are valuable and very helpful to revising and improving our manuscript. In what follows, we have improved the quality of the figures throughout the manuscript.
Reviewer 2 Report
I enjoyed reading this paper. A novel prokaryotic Argonaute family nuclease that shows promise for development of new tools for genome manipulation has been identified among Argonautes of mesophilic bacteria. The background survey of the study is adequate. The properties of the enzyme have been characterized by fairly extensive kinetic measurements. The experiments are described in sufficient detail. The presentation of results is systematic and easy to follow. The conclusions appear justified. The subject of this reseach is timely and probably of interest for quite an extensive readership.
Author Response
Thank you very much for your kind comments on our manuscript, and we really appreciate your valuable and helpful comments to improve our manuscript.
Reviewer 3 Report
Review on Ms. biomolecules-1533238
Jiang et al. analyzed the molecular properties of recombinant bacterial Argonaute (pAgo) proteins, with a particular focus on RsuAgo from Rummeliibacillus suwonensis (phylum Firmicutes). The latter is the only pAgo among the four analyzed proteins that was shown to be able to cleave RNA:DNA guide-target duplexes; pAgos are promising molecular tools, e.g. for the site-specific generation of DNA/RNA fragments in nucleic acid biochemistry and biology. The authors analyzed multiple aspects of RsuAgo function, including metal ion dependence, incubation temperature, guide:target mismatch sensitivity, dependence of plasmid cleavage on [NaCl] and G,C content). Basically, most of the experimental results are sound, demonstrating that mesophilic pAgos such as RsuAgo can be tailored to site-specific cleavage of DNA and RNA substrates, with several adjusting screws to increase cleavage efficiency and specificity. Unfortunately, several experimental results are neither adequately described nor correctly interpreted. Finally, the manuscript is hard to read and understand in its present form and thus requires extensive editing of English language and style.
Major specific comments:
1) Fig. 3, text and line 248-258: the authors speak about a burst phase of cleavage in the case of P-gDNA:tDNA and OH-gDNA:tDNA duplexes. I agree that this may be suggested by the curves shown in Fig. 3A/B, but this evidence would be strengthened if the authors fit the data to a single exponential and, for comparison, to a double exponential (see example in the attached file); the authors' statement would gain support if the fit were better with a double exponential decay equation. Also, I do not agree to describe the black and blue curves (P-gDNA: tRNA, P-gRNA: tDNA) in Fig. 3A as 'linear steady-state kinetics of target cleavage'. All one can infer from these two curves is that the buildup of a catalytic ternary complex (pAgo:guide:target) is much slower and rate-limiting in these cases.
2) Lines 276-278: the authors state here "RsuAgo-mediated tDNA cleavage guided by P-gDNA seemed not to show 5'-nucleotide preference"; I infer from Fig. 3B that a purine (A or G) is preferred over pyrimidines, and a 5'-C results in slowest kinetics. Please rewrite. Likewise, in lines 278-280, you may better say that the kinetics of P-gDNA-mediated tRNA cleavage are the fastest with 5'-A and the slowest with 5'-G (Fig. 3C). After saying that you may continue with the sentence "Despite these preferences, RsuAgo-mediated tRNA cleavage still maintains activity guided by gDNAs with 5'-C, G or T residues. In summary, the 2nd paragraph on p. 7 needs to be fundamentally revised. Why do the authors want to stress the point that the guide 5'-end nucleotide identity is not important? I rather think that the clear effect of guide 5'-end nucleotide identity on RsuAgo activity is a novel finding of this study, as it gives other researchers advice how to optimize pAgo cleavage for their specific target.
3) Line 288: Do you mean by ‘U-gRNA’ a P-gRNA with U at the 5’-end? If yes, why was it inactive compared with the 5’-U variant analyzed in Fig. 3D?
Line 294: gRNA with 5’-T or 5’-U?
4) Lines 305-307: the authors state that “… titration of Mn2+ ions showed that RsuAgo was equally active between 0.05 and 10 mM Mn2+ for DNA cleavage mediated by P-gDNA or OH-gDNA (Figure S7A and S7B), …” I infer from Fig. S7A and S7B that OH-gDNA mediates cleavage with decreasing efficiency at ≤ 1 mM Mn2+, whereas cleavage with P-gDNA is quite constant from 0.05 to 10 mM Mn2+.
5) Line 322, Fig. 4C: I do not see a ‘burst phase’ with Mn2+ in Fig. 4C.
Line 325: what is meant by ‘from a linear growth trend’ ?
6) Fig. S8B: why are the 45 nt tRNA and 34 nt product bands are so faint? In lines 344/345 the authors say that RNA cleavage was stimulated at 45°C (illustrated in Fig. 5A), but in Fig. S8B, the intensity ratios of the 45 and 34 nt bands appear to be quite similar between 30 and 65°C. Please explain this discrepancy.
7) Line 364 “We found that 14-40 nt P-gDNA were sufficient for RsuAgo-mediated DNA target cleavage (Figure 5B, Figure S8C and S8D), …” I see in Fig. S8C that cleavage largely drops at a gDNA length of 30 nt and is essentially not detectable with gDNA of 40 nt (taking into account that the lane is overloaded); this does not correpond to equal heights for the red columns representing the 30 and 40 nt long gDNAs in Fig. 5B. Furthermore, with a 14-nt gDNA there is essentially no cleavage (Fig. S8C); thus, I see a length window for efficient cleavage from 15 nt to 21 or possibly 25 nt.
8) paragraph 3.7, beginning in line 378: it would be helpful to illustrate the mismatching duplexes of the 18-nt gDNA with tDNA and tRNA, such that the reader can see the type of mismatch pairs (purine:purine, pyrimidine:pyrimidine).
9) LIne 418: this part sentence is cryptic.
Please also see the file in the pdf version with some added comments.

Author Response
We appreciate you very much for the constructive comments and suggestions on our manuscript. Again, we appreciate your time and efforts in helping us improve our work.
Please see the attachment of detailed response.

Round 2
Reviewer 3 Report
Review on Ms. biomolecules-1533238, version 2
I have only two remaining major points and minor text corrections (sorted by pages) to be considered before publication.
1) There was a misunderstanding regarding my comment 1 (Figure 3). What I meant pertains to the time course curves for P-gDNA:tDNA in Fig. 3A and for 5'-A P-gDNA:tDNA; there it looks like that the first data point (about 22% cleaved) suggests a steeper initial increase relative to the overall curve shape resulting from fitting to a single exponential decay equation. This suggests that a fraction (about 22%) of preformed RsuAgo:gDNA:DNA target complex reacts particularly fast, but then the rate becomes somewhat slower; here, the data might better fit a double exponential decay equation. If so, then this initial phase (up to 22% cleaved) has been termed "initial burst" in enzyme kinetics.
However, this issue is not so important in the context of your study and some of the curve fits in the revised Figure 3 are suboptimal (for example, the fits of the two lower curves in panel A and those in panel C). So I propose that you better keep the original colored Figure 3 where the data points are simply connected by lines. Sorry for the confusion.
2) In the new Figure 5 B and D, as in Figure S7F, there seems to be RNA degradation in the presence of 5 mM Mn2+. RNA degradation is not apparent in Figure 4B, why?
Since site-specific RNA cleavage by RsuAgo is one of the most interesting applications, do the authors have any ideas how to reduce or prevent degradation of RNA targets (e.g. by pH reduction or by lowering the Mn2+ concentration)? This aspect should be discussed in the manuscript.
3) Please indicate in the legend how long reactions were incubated in Fig. 4A/B.
Suggestions for text corrections:
(line numbers refer to the file ' biomolecules-1533238-peer-review-v2.pdf')
Page 1:
- line 65, change to: "... RsuAgo-mediated RNA target cleavage occurs only with short guide DNAs in a narrow length range (16-20 nt), and mismatches ..."
- line 67, change to: "RsuAgo-mediated target cleavage shows a preference for a guide strand with a 5'-terminal A residue. Furthermore, we have have found that RsuAgo can cleave double-stranded DNA ..."
- line 89: "... pAgos with their powerful capacities may as well become valuable tools for applications in biotechnology [10-13]."
Page 2:
- line 153: "... RNA, and in view of the SARS-CoV-2 pandemic [20], it is mandatory to further explore RNA cleavage ..."
- line 156: delete "thus"
- line 158: "... Considering the potential application of mesophilic pAgos in genome editing and RNA manipulation, we proceeded here to identify ..."
- line 163: "... RsuAgo-mediated RNA target cleavage occurs with only ..."
- line 166: "... about 45% of superhelical dsDNA ..."
- line 168: "... pAgos-based ..."
- line 179: "... Mutations were introduced into RsuAgo ..."
Page 3:
- line 206: "For RsuAgo, these fractions were further loaded onto ..."
- line 215: "... to 100 pmol / uL in DEPC (diethylpyrocarbonate)-treated water and 5'-phosphorylated with T4 PNK (T4 polynucleotide kinase); the ..."
- line 224: "... and used the web-based BLASTp algorithm of the NCBI database ..."
- line 226: "The evolutionary relationship between the four pAgos and several other characterized pAgos was analyzed with the MEGA 7.0 software [22]."
- line 235: "... 200 nM target were added. After reaction for the indicated time ..."
- line 239: "... (PAGE) and stained ..."
- line 241: "... at the 5'-end ..."
- line 242: "... and the kinetics of cleavage were monitored over time. Thereafter, the denaturing PAGE gels ..."
- line 246: "To test whether the four pAgos act uniformly on DNA and RNA targets ..."
- line 247: "... analyze cleavage activity. gDNA or gRNA was loaded onto pAgos, and then the Ago-guide complex ..."
Page 4:
- line 297: "... kinetics were tested ..."
- line 298: "... containing a 5'-A, 5'-C, 5'-G, or 5'-T (5'-U) residue at the 5'-end."
- line 314: "... single nucleotide mismatches in the 5'-A-gDNA ..."
- line 321: "... glycerol) supplemented with 5 mM NaCl if not stated otherwise. The two reaction mixtures were ..."
- line 322: "... 200 ng of plasmid DNA were ..."
- line 325: "... in this study are listed in Supplementary Table S2. When exploring cleavage of plasmid DNA as a function of NaCl concentration, considering that the storage buffer of RsuAgo protein contained 500 mM NaCl, ..."
- line 336: "... identity (about 50%) and aligned with KmAgo. Phylogenetic analysis revealed that RsuAgo, BsAgo and DeAgo are closely related to KmAgo, while PmAgo is more distantly related to the former three pAgos (Figure 1A)."
- line 339: "... RsuAgo, BsAgo, DeAgo and PmAgo contain a conserved catalytic ..."
Page 5:
- line 382: "... consistent with the predicted molecular weights (Figure S2-S4). Then, to confirm whether RsuAgo, BsAgo, DeAgo and PmAgo are indeed active nucleases, we performed ..."
- line 386: "... was added, followed by incubation at 37 °C ..."
- line 389: "For PmAgo, no cleavage activity at all was observed ..."
- line 390: "... guided by P-gDNA, but also cleaved DNA ..."
Page 6:
- line 432: "Figure 2. Analysis of single-stranded ..."
- line 433: "Cleavage by RsuAgo ..."
- line 436: "... Watson-Crick pairing. F-M1 (FAM-M1) and F-M2 (FAM-M2) are chemically synthesized, 34 nt long ssDNA (F-M1) and ssRNA (F-M2) oligonucleotides with a FAM-label at the 5'-end that were loaded on gels as size markers corresponding to the length of the DNA and RNA cleavage products, respectively."
- line 444: "... synthesized 34 nt long FAM-labeled ssDNA or ssRNA as markers, and their sequences corresponded to the predicted ..."
- line 447: "... 34 nt long 5'-fragment ..."
- line 458: "3.3. Kinetic analysis of single-stranded nucleic acid cleavage by RsuAgo"
- line 462: "... (RsuAgo-guide complex in excess over tDNA) ..."
- line 463: see comment 1 above.
Page 7:
- line 485: "... started with a burst phase ..."
Page 8:
- line 541: "... containing 5 mM Mn2+ ions ..."
- line 545: "Previous pAgo studies reported ..."
- line 547: "... [4, 18, 28, 29]. In addition, certain eAgos and pAgos showed a bias for a specific guide 5'-nucleotide [19]."
- line 549: "... for the 5'-nucleotide of gDNA or gRNA, we ..."
- line 550: "According to time course analysis of tDNA cleavage, ..."
Page 9:
- line 621: "... resulted in substantial tDNA cleavage activity (Figure 3B, 3D)."
- line 629: "... we performed a 1 h cleavage reaction ..."
- line 634: "... gRNA-programmed RsuAgo could perform ..."
- line 635: "... but its cleavage activity was poor (Figure 3D)."
- line 638: "... As expected, gRNA-mediated cleavage efficiency was significantly
lower than that mediated by gDNA. Moreover, there is only a weak base preference for the 5'-terminal gRNA nucleotide in gRNA-mediated tDNA cleavage.”
- line 646: "... for pAgo activity …”
- line 647: "... cations on the kinetics of guide-dependent tDNA or tRNA cleavage.”
- line 655: "... In contrast to pAgo-mediated RNA cleavage by KmAgo [17], RsuAgo was unable to use OH-gDNA as a guide for RNA cleavage. Also, P-gDNA-mediated RNA cleavage experiments revealed differences in the utilization of Mn2+ and Mg2+ ions. RsuAgo cleaved RNA only at Mg2+ ions concentrations above 1 mM (Figure S7E), while it could perform RNA cleavage at Mn2+ ions concentrations ranging from 0.05 mM to 10 mM, with comparable efficiency (Figure S7F).”
- line 664: "... in the presence of Mn2+, but was much less active with …”
- line 665: "And no cleavage activity was observed for both substrates, tDNA and tRNA, in the presence of Ca2+, Co2+, Ni2+, Cu2+ or Zn2+.”
- line 670: "... increase phase under Mg2+ conditions. Thus, RsuAgo-mediated cleavage in the Mn2+ buffer was more efficient than in the Mg2+ buffer.”
- line 672: "... we observed that the Mn2+-assisted cleavage reaction started with a lag phase
(Figure 4D) and this lag phase was extended from ~10 to 30 min in Mg2+ buffer. We propose that the lag phase reflects slow build-up of the ternary and/or catalytically active complex in RsuAgo-mediated RNA cleavage [17].”
Page 11:
- line 721: "... in triplicates and quantified. Error bars represent the SD based on three independent experiments.”
- line 726: "Analysis of tDNA cleavage activity at different temperatures showed …”
- line 729: "… significantly less effective than DNA cleavage at most temperatures, and the RNA target and product bands were rather faint. We speculate that the degradation was caused by Mn2+, since the RNA target and product signals decreased with increasing Mn2+ concentration (Figure S7F).”
Page 12:
- line 745: "... Effects of P-gDNA length on DNA …”
- line 746: "... Effects of P-gDNA length on RNA …”
- line 748: "... 5 mM Mn2+ ions for 1 h at 37 °C.”
Page 13:
- line 791: "... series of P-gDNAs from 9 …”
- line 793: "... 15-25 nt long P-gDNAs were …”
- line 795: "... on CbAgo showed that it can bind a single-stranded guide …”
- line 797: "... tDNA in the presence of gDNA with …”
- line 799: "... KmAgo [17], RsuAgo cleaved RNA only with short 16-20 nt P-gDNAs …”
- line 800: "... extended guide length essentially abolished cleavage activity. It is known that FpAgo promotes DNA cleavage only with short 15-20 nt gDNAs, rather than tolerating a wide range of guide lengths [32], but restrictions in the guide length for RNA cleavage by pAgos have not yet been reported.”
- lines 804/805: delete this sentence (has been said before)
- line 808: "... that Agos identify their target via a guide oligonucleotide that forms Watson-Crick base pairs with the target, and mismatches …”
- line 813: "... [14, 17, 18, 30]. Here we explored …”
Page 14:
- line 876: "... (blue) and DNA/ RNA target (grey). The different functional gDNA elements are indicated above the gDNA; sup., supplementary region.”
- line 879: "Control, gDNA with full complementarity to the target. All experiments were …”
- line 881: "... in reaction buffer containing 5 mM Mn2+ …”
- line 882: All cleavage experiments were carried out in triplicates and quantified. Error bars represent the SD based on three independent experiments.”
- line 887: "... did not impair RsuAgo-mediated cleavage activity, but rather stimulated tRNA cleavage …”
- line 889: "... was extremely sensitive to single nt mismatches in the seed region (m4, m5, m7, m8), central region (m9, m11, m12) and supplementary region (m14, m15). Thus, mutation at most positions dramatically reduced or nearly abolished RNA cleavage activity, with the exception of 6 positions (m6, m10, m13 and m16-m18 in the 3'-tail region [18]) (Figure 6C, D and Figure S8B).
- line 893, rewrite: "In comparison, mismatches at m7, m8, and m14 significantly reduced target RNA cleavage and those at m9-m13 almost eliminated RNA cleavage activity catalyzed by KpAgo from the yeast Kluyveromyces polysporus [33]. In the case of KmAgo, dinucleotide mismatches m11/m12 (tDNA cleavage) and m8/m9 (tRNA cleavage) caused the strongest decreases in cleavage efficiency [17].
- line 899: "... Here we have characterized, for the first time, RsuAgo ..."
- line 900: "... 2 sites, and catalyzed DNA-guided ..."
- line 901: "... 9 sites of the gDNA. We further showed that introduction of single nt mismatches (m2, m3) into the seed region enhances RNA cleavage and mismatches at 9 other positions eliminate RNA cleavage activity, which provides important information for applications of RsuAgo in RNA manipulation. Likewise, for mouse Ago2 intentionally weakened seed pairing enhanced cleavage activity relative to the guide-target duplex with full complementarity [25, 35]."
- line 920: "... increased DNA strand separation [19, 36] such that pAgos can effectively ..."
Page 15:
- line 949: "... nicked the plasmid DNA (Figure 7B). Only when supplied with a pair ..."
- line 952: "... content of 45% or lower (Figure 7B), but not (or barely) regions with higher GC-content (Figure 0S9). These results demonstrate that RsuAgo is able to mediate DNA-guided cleavage of plasmid dsDNA, the efficiency depending on NaCl and GC-content of the dsDNA target."
Page 16:
- line 957: "Figure 7. Cleavage of double-stranded plasmid DNA by RsuAgo. (A) RsuAgo-mediated cleavage of plasmid pUC19 at different NaCl concentrations. RsuAgo and a gDNA with 29% GC-content were preincubated for 30 min at 37 °C, followed by addition of pUC19 dsDNA, incubation for 2 h at 37°C in in reaction buffer containing 0.5 mM Mn2+ and cleavage analysis by 1% agarose gel electrophoresis.
(B) RsuAgo-mediated pUC19 cleavage with gDNAs differing in their GC-content in reaction buffer containing 0.5 mM Mn2+ and ?? NaCl for 2 h at 37 °C. ScaI lane: the isolated
plasmid was digested with ScaI for 2 h. FW/RV-gDNA, forward and reverse gDNA
corresponding to a specific target site in plasmid pUC19; NC-gDNA, a non-complementary gDNA control; M, molecular weight marker; ..."
- line 970: "... As KmAgo ..."
- line 971: "... 50% sequence identity to KmAgo ..."
- line 975: "... In addition, RsuAgo preferred a guide with a 5'-A residue, but this bias was moderate, since guides with a 5'-C, 5'-T or 5'-G residue gave also rise to substantial target cleavage."
- line 980: "RsuAgo could cleave DNA with gDNAs in the length range of 15-25 nt. However, in the case of RNA cleavage, the length range of gDNAs mediating RNA cleavage was confined to 16-20 nt. Moreover, RsuAgo showed a strong mismatch sensitivity in gDNA-mediated RNA cleavage. Mismatches at 9 single positions of the guide-target duplex strongly reduced or even eliminated cleavage by RsuAgo."
Page 17:
- line 1040: "... dsDNA plasmid linearization at 37 °C and was able to cleave dsDNA target sites with up ..."

Author Response
Response to Reviewer 3 Comments
Point 1: 1) There was a misunderstanding regarding my comment 1 (Figure 3). What I meant pertains to the time course curves for P-gDNA:tDNA in Fig. 3A and for 5'-A P-gDNA:tDNA; there it looks like that the first data point (about 22% cleaved) suggests a steeper initial increase relative to the overall curve shape resulting from fitting to a single exponential decay equation. This suggests that a fraction (about 22%) of preformed RsuAgo:gDNA:DNA target complex reacts particularly fast, but then the rate becomes somewhat slower; here, the data might better fit a double exponential decay equation. If so, then this initial phase (up to 22% cleaved) has been termed "initial burst" in enzyme kinetics.
However, this issue is not so important in the context of your study and some of the curve fits in the revised Figure 3 are suboptimal (for example, the fits of the two lower curves in panel A and those in panel C). So I propose that you better keep the original colored Figure 3 where the data points are simply connected by lines. Sorry for the confusion.
Response 1: We are very sorry for our misunderstanding regarding your comment 1, and we are very grateful for your suggestion. We have kept the original colored Figure 3.
Point 2: 2) In the new Figure 5 B and D, as in Figure S7F, there seems to be RNA degradation in the presence of 5 mM Mn2+. RNA degradation is not apparent in Figure 4B, why? Since site-specific RNA cleavage by RsuAgo is one of the most interesting applications, do the authors have any ideas how to reduce or prevent degradation of RNA targets (e.g. by pH reduction or by lowering the Mn2+ concentration)? This aspect should be discussed in the manuscript.
Response 2: We are very sorry for no additional explanation, something is misleading about Figure 4B. The gel (Figure 4B) was stained with SYBR Gold longer than the gel (Figure 5B and Figure S7F). This made the target and product bands were very visible without a faint situation.
We are very grateful for your suggestion, it is necessary to prevent the degradation of RNA targets. Thus, we have modified the discussion. Specific modifications are listed below.
“and the RNA target and product bands were rather faint. We speculate that the degradation was caused by Mn2+, since the RNA target and product signals decreased with increasing Mn2+ concentration (Figure S7F). Considering the practical application of RNA manipulation in the future, we can reduce the Mn2+ concentration to avoid the degradation of the target and product bands in the later research, and we need to further explore other pathways to prevent degradation of RNA targets.”
Point 3: 3) Please indicate in the legend how long reactions were incubated in Fig. 4A/B.
Response 3: We are very grateful for your suggestion. Specific modifications are listed below.
Line 346-348: “Considering that divalent metal ions are important cofactors for pAgo activity, we investigated the effects of divalent metal cations on the kinetics of guide-dependent tDNA or tRNA cleavage for 1 h at 37 °C.”
Line 394-396: “All experiments were performed at the 4: 2: 1 RsuAgo: guide: target molar ratio in reaction buffer containing different divalent metal ions for 1 h at 37 °C.”
In the end, we are very grateful for your suggestion, and we have indicated reaction time throughout the full-text content.
Point 4: Suggestions for text corrections: (line numbers refer to the file ' biomolecules-1533238-peer-review-v2.pdf')
Response 4: Thanks a lot for your suggestions. We have corrected the corresponding text according to your suggestions.
In the end, we appreciate the reviewer very much for your constructive comments and suggestions on our manuscript. Again, we appreciate your time and efforts in helping us improve our work.
We appreciate the reviewer very much for your constructive comments and suggestions on our manuscript. Again, we appreciate your time and efforts in helping us improve our work.